# Field Evaluation of the Performance of Two Rapid Diagnostic Tests for Meningitis in Niger and Burkina Faso

**DOI:** 10.3390/microorganisms9040832

**Published:** 2021-04-14

**Authors:** Marc Rondy, Mamadou Tamboura, Fati Sidikou, Issaka Yameogo, Kambire Dinanibe, Guetwende Sawadogo, Chantal Kambire, Halima Mainassara, Ali Elhaj Mahamane, Baruani Bienvenu, Haladou Moussa, Rasmata Ouedraogo, Katya Fernandez, Muhamed-Kheir Taha, Olivier Ronveaux

**Affiliations:** 1Family, Gender and Life Course Department, Comprehensive Family Immunization Unit, Pan American Health Organization, Washington, DC 20037, USA; rondym@who.int; 2Centre Hospitalier Universitaire Pédiatrique-Charles De Gaulle, Ouagadougou 01 BP 1198, Burkina Faso; matouboura@yahoo.fr (M.T.); ramaouedtra@yahoo.fr (R.O.); 3Centre de Recherche Médicale et Sanitaire (CERMES), Niamey 10887, Niger; fatsidik@yahoo.fr (F.S.); halima@cermes.org (H.M.); elmahamane@cermes.org (A.E.M.); 4Direction de la Protection de la Santé de la Population, Ouagadougou 03 BP 7035, Burkina Faso; yameogoissaka@yahoo.fr; 5Unité VIH et Santé de la Reproduction (UVSR), Ouagadougou 03 BP 7192, Burkina Faso; dinanibekambire@yahoo.fr; 6Davycas International, Ouagadougou 08 BP 11593, Burkina Faso; guetasawadogo@gmail.com; 7World Health Organization, Burkina Faso Office, Ouagadougou 03 BP 7019, Burkina Faso; kambirec@who.int; 8World Health Organization, Niger Office, Niamey B.P. 10739, Niger; baruaningoyb@who.int (B.B.); haladoum@who.int (H.M.); 9World Health Organization, CH-1211 Geneva, Switzerland; 10Institut Pasteur, 75015 Paris, France; muhamed-kheir.taha@pasteur.fr; 11Pan American Health Organization, Dominican Republic Office, Santo Domingo 1464, Dominican Republic; ronveauoli@paho.org

**Keywords:** meningitis, *Neisseria meningitidis*, *Streptococcus pneumoniae*, rapid diagnostic test, national reference laboratory, cerebrospinal fluid, Niger, Burkina Faso

## Abstract

New lateral flow tests for the diagnosis of *Neisseria meningitidis* (Nm) (serogroups A, C, W, X, and Y), MeningoSpeed, and *Streptococcus pneumoniae* (Sp), PneumoSpeed, developed to support rapid outbreak detection in Africa, have shown good performance under laboratory conditions. We conducted an independent evaluation of both tests under field conditions in Burkina Faso and Niger, in 2018–2019. The tests were performed in the cerebrospinal fluid of suspected meningitis cases from health centers in alert districts and compared to reverse transcription polymerase chain reaction tests performed at national reference laboratories (NRLs). Health staff were interviewed about feasibility. A total of 327 cases were tested at the NRLs, with 26% confirmed Nm (NmC 63% and NmX 37%) and 8% Sp. Sensitivity and specificity were, respectively, 95% (95% CI: 89–99) and 90% (95% CI: 86–94) for Nm and 92% (95% CI: 75–99) and 99% (95% CI: 97–100) for Sp. Positive and negative predictive values were, respectively, 77% (95% CI: 68–85) and 98% (95% CI: 95–100) for Nm and 86% (95% CI: 67–96) and 99% (95% CI: 98–100) for Sp. Concordance showed 82% agreement for Nm and 97% for Sp. Interviewed staff evaluated the tests as easy to use and to interpret and were confident in their readings. Results suggest overall good performance of both tests and potential usefulness in meningitis outbreak detection.

## 1. Introduction

Because of its high case fatality (around 10% [1,2]) and epidemic potential, meningococcal meningitis is a major public health threat [3], especially in the “meningitis belt”. This region, which stretches from Senegal to Ethiopia, is characterized by a high seasonal incidence of meningococcal meningitis between January and June [4].

Meningococcal meningitis caused by *Neisseria meningitidis* (Nm) bacteria may cause large outbreaks and six of the 12 Nm serogroups (A, B, C, W, X, and Y) are responsible for invasive forms of meningitis [5]. The epidemiology of Nm meningitis is evolving rapidly, and new variants are continually emerging [6]. 

A wide choice of meningococcal vaccines covering a variable number of serogroups is currently available [7]. In a context of a diversification of serogroups with epidemic potential in the meningitis belt, it is important to strengthen microbiological surveillance to prompt the early implementation of vaccination campaigns appropriate to circulating serogroups.

Pneumococcal meningitis causes clusters of cases and, less frequently, outbreaks, in the meningitis belt, even following introduction of pneumococcal conjugate vaccines. In 2019, following massive the reduction of NmA, with MenA vaccine rollout, *Streptococcus pneumoniae* (Sp) was found to be responsible for 40% of meningitis cases in countries in the region [8]. Pneumococcal meningitis has very high case fatality rates (36–66% in the meningitis belt [9]) and, given the difficulty of treatment, requires longer treatment protocols than those for meningococcal meningitis [10]. 

The microbiological diagnosis of meningitis is currently based on culture or, more frequently, on polymerase chain reaction (PCR) tests carried out by national reference laboratories (NRLs) on a sample of cerebrospinal fluid (CSF) [11]. This last procedure, which requires maintaining the sample in a cold chain in primary health care centers (PHCs) and during transport to the NRL, makes the early identification of circulating strains difficult. To reduce the time needed to detect circulating strains, diagnostic tests have been developed that can be used at bedside and provide a diagnosis in a few minutes. However, currently available rapid diagnostic tests (RDTs) to detect meningococcal meningitis have a short shelf life, are sensitive to heat, and cannot detect all circulating serogroups, making them of little utility in the meningitis belt context [12]. 

The rapid diagnostic tests (RDT) were first developed at the Institut Pasteur, Paris, as immunochromatographic tests [13,14]. They were thereafter transferred the BioSpeedia company who then developed MeningoSpeed and PneumoSpeed that can be used between 2 and 30 °C to diagnose, respectively, the five main meningococcal serogroups (A, C, W, X and Y) [15] found in the African meningitis belt and Sp meningitis. The tests are based on the use of antibodies directed against capsular polysaccharide of Nm (serogroups A, C, W, Y, and X) and against pneumococcal cell wall polysaccharide C that is common to all Sp isolates. Laboratory based tests suggest good sensitivities and specificities of these two RDTs [12].

These RDTs require the transfer of two drops of CSF in each cassette (three for Nm groups in A/W serogroups, Y/C serogroups and X serogroup; one for Sp meningitis) and a waiting time of 15 min. The reading process is similar to any RDTs, based on control and case lines appearance and they can be performed by all staff after a short demonstration.

The objective of the study was to measure the performance of the MeningoSpeed and PneumoSpeed RDTs in diagnosing Nm and Sp meningitis in CSF at the bedside among patients in PHCs in Niger and Burkina Faso in 2018 and 2019. 

## 2. Materials and Methods

### 2.1. Overview 

To measure the performance of the RDTs, we compared the results of RDTs conducted at bedside in PHCs with the results of PCRs obtained at NRLs. PHCs in districts where incidence exceeded 3 suspected cases per week per 100,000 inhabitants were invited to participate, and staff were trained, supervised and supplied with RDTs. 

To measure the performance of the RDTs after sample transportation, RDTs were also repeated at the NRLs and results compared with PCR tests. 

The study population consisted of any consenting patient aged at least two months admitted to a participating PHC with suspected bacterial meningitis (according to the WHO case definition [16]) (sudden onset of fever and stiff neck or other meningeal signs (including bulging fontanelle for patients under 12 months)) between April 2018 and June 2019. Suspected cases with contraindications for lumbar puncture or who refused to participate were excluded from the study.

According to national guidelines for suspected bacterial meningitis case management, medical staff collected CSF through lumbar puncture on every suspected case. For each patient included in the study, the two RDTs (MeningoSpeed and PneumoSpeed) were performed on CSF at the PHC. A 1 mL CSF tube was then refrigerated and sent to the NRL where an additional RDT and a real-time PCR test was performed and used as a gold standard.

PHC staff collected information on patients’ symptoms; antibiotic treatment before admission; dates of onset of symptoms, lumbar puncture, RDT reading, and dispatch of sample to the NRL; lot number; and RDT results. PHC information was merged with NRL data, including the results of the PCR tests and the RDTs performed at the NRLs, and data were analyzed using the STATA software package version 14 [17]). The producer of the tests (BioSpeedia Company, Saint Etienne, France) did not fund the trail, did not contribute to the design and did not participate in the analysis of the results. None of the authors is employed by that company.

### 2.2. Performance of the RDTs

MeningoSpeed test results were classified as positive or negative per serogroup A, C, W, Y, or X (positive if positive for that serogroup; negative if negative for that serogroup, regardless of other serogroup’s results) and for any Nm (positive if positive for any serogroup A, C, W, Y, or X; negative if negative for all these 5 serogroups). PneumoSpeed test results were classified as positive or negative for Sp meningitis.

We compared the results of the RDTs performed in the PHCs with the results of PCR tests performed at the NRLs to calculate the sensitivity, specificity, positive predictive value (PPV) and negative predictive value (NPV) of the tests. The analyses were stratified by year of lumbar puncture (2018 vs. 2019) and epidemic period (defined as between January and June) versus non-epidemic period.

### 2.3. RDT Feasibility and Acceptability

PHC staff photographed the RDTs as they were reading them. The photographs were then reviewed and the RDTs blindly re-interpreted by independent readers. We compared the results of the RDTs as interpreted by the PHC staff and independent experts and quantified the level of concordance between them using the kappa coefficient [18]. We considered the agreement between the tests moderate if κ < 0.4, average or good if κ = 0.4 to 0.75 and excellent if κ > 0.75. National study coordinators conducted semi-structured interviews with RDT users to assess the acceptability and feasibility of the RDTs.

### 2.4. Sample Size

To estimate an expected 95% sensitivity of MeningoSpeed Nm meningitis identification, with an absolute precision of 5% and a 30% PCR positivity [19], we estimated that at least 243 suspected cases should be included in the study.

### 2.5. Ethical Considerations

The study followed the principles governing biomedical research involving human participation and was carried out in line with principles of the Declaration of Helsinki to ensure that the rights, integrity, and confidentiality of participants were protected. The agreements of the ethics committees of WHO (WHO ERC.0002926), Niger (deliberation N° 35/2017/CNRES) and Burkina Faso (deliberation N° 2017-10-156) were obtained before the start of the study as well as for its extension into 2019.

Written consent was obtained from suspected meningitis cases or from their parents or legal guardians. Standardized information was read to the potential participant. The confidentiality of the data collected, and the anonymity of the participants were ensured during and after the survey (no patient name appears in the databases).

## 3. Results

### 3.1. RDT Performance

Between 8 April 2018 and 30 June 2019, 421 people with suspected meningitis were admitted to the participating PHCs and tested with the RDTs. Of those people, 327 were eligible for the study, with completed questionnaires and PCR results entered into the database (246 in Niger, and 81 in Burkina Faso) and 198 (61%) were recruited during the epidemic period. The distribution of symptoms among suspected cases was typical of that found in the meningitis belt. A total of 106 (32%) and 28 (9%) patients tested positive for Nm and Sp, respectively, with the RDTs in the PHCs (Table 1).

Of the 327 cases with PCR results obtained at an NRL, 86 (26%) were confirmed with Nm (NmC 63% and NmX 37%) and 26 (8%) with Sp (Table 2). The median time between lumbar puncture and PCR result was 18 days outside the epidemic period, and 22 days during the epidemic period.

Sensitivity and specificity were, respectively, 95% (95% CI: 89–99) and 90% (95% CI: 86–94) for any Nm, 93% (95% CI: 82–98), and 98% (95% CI: 95–99) for NmC, and 91% (95% CI: 75–98) and 96% (95% CI: 93–98) for NmX.

PPV and NPV were, respectively, 77% (95% CI: 68–85) and 98% (95% CI: 95–100) for any Nm. PPV was better in 2019 than in 2018: 90% (95% CI: 81–96) vs. 49% (95% CI: 31–67), *p*-value < 0.01 (Table 3); and was better during epidemic months: 89% (95% CI: 79–95) versus 53% (95% CI: 35–70), *p*-value < 0.01 (Table 3 and Table 4).

For Sp, sensitivity and specificity were, respectively, 92% (95% CI: 75–99) and 99% (95% CI: 97–100), and PPV and NPV values were 86% (95% CI: 67–96) and 99% (95% CI: 98–100) (Table 2).

Owing to a lack of tests and a strike by staff, it was only possible to repeat the RDTs at the NRLs in 147 of the 327 patients (45%). Performances of the RDTs at the NRLs and the health centers were comparable (Table 2).

### 3.2. RDT Feasibility and Acceptability

There was an 82% agreement for Nm and 97% for Sp between RDT results registered at the PHCs and photographs reviewed by an independent expert. The kappa coefficient suggested an excellent concordance for Sp and NmX, a good concordance for all Nm and moderate concordance for NmA, NmC, and NmW (Table 5).

Eleven NmX, nine NmA, six NmC, two NmW, and one NmY were detected positive with RDTs but were tested negative by real-time PCR.

All nine positive NmA results identified in health centers were from Burkina Faso, eight in 2018 and one in 2019. Of these, two were identified as positive for NmA on photographs by the external expert. The authors confirmed that a faint line, suggestive of a positive NmA result was indeed visible in these two photographs (Figure 1). Of the eleven false positive NmX results, nine were from Burkina Faso and two from Niger. Of those, four photographs were reviewed and classified as negative by the independent experts. One photograph of a false positive NmC result was available and suggested a misreading by the PHC medical staff.

A total of 31 staff were interviewed about their experiences in using the RDTs (20 in Niger and 11 in Burkina Faso). Overall, respondents found the RDTs easy to use (average difficulty score: 1.9/10). They found their interpretation very easy (average difficulty score: 1.2/10) and had great confidence in their result (average lack of confidence score: 1.3/10).

## 4. Discussion

### Performance of the RDTs

We described in this article the performance results from ready-to-use bedside RDTs, usable by any medical staff, storable at 2–30 °C, allowing to detect Nm and Sp within 15 min. Our results suggest that the RDTs for diagnosing Nm and Sp performed well under field conditions at PHC level, with sensitivity and specificity above 90%. They performed similarly whether used by medical staff at a PHC or technical staff at an NRL. These RDTs were described as acceptable and easy to use in a bedside context. Serogroup-specific Nm RDT performance could be measured for NmC and NmX and were in the same range, above 90%. The two serogroups were the most prevalent in West Africa in 2018–2019 [20,21]. Further studies would be needed to validate their use in detecting NmA, NmW, and NmY due to low prevalence during our study period.

By extending the study over two years we were able to obtain a sample size large enough to measure the performances of the RDTs with acceptable precision. The study protocol was well followed, despite challenges in systematizing the capture of photographs and the use of the RDTs at the NRLs. In addition, recruitment was lower in Burkina Faso due to security issues along with a strike by health workers in 2019. Owing to these limitations, the precision of some secondary outcomes was low. Logistical issues between the PHCs and NRLs, potentially leading to alteration of samples and affecting the measured performance of the RDTs, cannot be excluded. In order to account for such a risk, the protocol included repetition of RDTs at the NRLs. Similar results between RDTs carried out at the PHCs and NRLs suggest that sample alteration was low in this study.

Our results suggest a higher sensitivity of the RDT in detecting NmC in this field context than previously reported in laboratory conditions (95% versus 65%) [15]. This could be due to the fact that the circulating clone in Niger and Burkina Faso (NmC, clonal complex ST-10217) [19,22] is particularly well detected by the antibodies used in the MeningoSpeed test.

PPV for any Nm and NmX were at 77% and 73%, respectively, corresponding to a quarter of false positive RDT readings for these outcomes. Moreover, recurrent false positives for NmA were registered. Further analyses suggested that most of these findings were attributed to specific PHCs and that constant supervision and training of RDT users led to a significant decrease in false positives between the first and the second study years. NmA RDT issues, documented with photographs, were fed back to the manufacturer.

The original study protocol included the use of PneumoSpeed tests on urine samples. However, only 31 tests were reported, precluding any interpretation related to this objective. Considering its potential impact on patients’ comfort and practice safety, specific studies should be implemented to measure the performance of RDT tests on urine.

Performance measurements are the results of both the quality of the test and the interpretive capacity of the user. Measurement of the concordance between the interpretations of the nursing staff and the independent expert made it possible to discuss this additional information bias linked to the evaluator. Although the proportion of photographs that could be read was low, concordance was generally good, suggesting a good understanding of the use of the RDTs by the PHC staff.

These field performance results met the WHO target product profile acceptable values for sensitivity and specificity, of >90% [23] and were superior or comparable to field performance values for existing meningococcal rapid tests (69–80% sensitivity and 81–94% specificity for latex agglutination tests and 89–92% sensitivity and 85–99 specificity for lateral flow test) [12]. Our field evaluation confirms the good performance of the tests in laboratory conditions and suggests that the tests are suitable for use in field conditions and that they are acceptable to health personnel, but that they should be accompanied by clear instructions and effective training. Considering this performance, their longer shelf-life and improved thermostability (but still below the desired target product profile value of 40°), easier test procedures, and inclusion of all main Nm serogroups (including NmX), MeningoSpeed and PneumoSpeed are good candidate tests for the early detection of meningitis epidemics in Africa. However, this field study is only one step towards ensuring access to safe appropriate diagnostic tests of good quality. At any rate, ensuring confirmation of test results with a more specific test such as PCR will continue to be key, given the decrease in incidence of Nm A and anticipated decrease of other Nm serogroups with future vaccination efforts.

One of the priority goals identified by the Defeating meningitis by 2030 Roadmap is the improvement of diagnosis at all levels of care, through the development and access to diagnostic assays [24]. An expert group gathered by WHO in 2018 [25] identified three essential objectives for the development of in vitro diagnostic tests (IVD) for meningitis diagnosis, including the rapid detection of epidemics in the African meningitis belt. The MeningoSpeed and PneumoSpeed could potentially meet this specific need. Issues that remain to be addressed, before procuring the test more widely, are costs, further thermostability improvements, and scale-up of production capacity with a reliable quality management system. Until elimination of meningitis epidemics in the region is achieved, one of three visionary goals of the roadmap, the use of rapid diagnostic tests will remain an important tool for meningitis control in Africa.

## Figures and Tables

**Figure 1 microorganisms-09-00832-f001:**
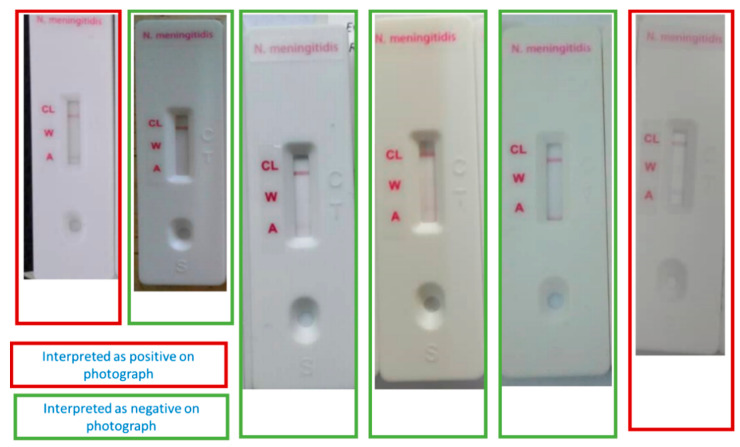
Photographs of six false positive NmA RDTs by independent expert classification, Burkina Faso and Niger, 2018–2019.

**Table 1 microorganisms-09-00832-t001:** Characteristics and distribution of symptoms among suspected cases (N = 327), Niger and Burkina Faso, 2018–2019.

	Characteristics	N	%
	Total patients	327	100
Recruitment country	Burkina Faso	81	25
	Niger	246	75
Recruitment period	Epidemic	198	61
	Non-epidemic	129	39
Age—average (range)		9 years (3 months—86 years)	
Symptoms	Sudden onset of fever	316	97
	Abdominal pain	181	55
	Confusion and disorientation	180	55
	Joint pain	136	42
	Stiff neck	116	36
	Bulging fontanelle	10	3
	Petechial rash	1	0
	other meningeal signs	18	6
Nm RDT results	All serogroups	106	32
	NmA	9	3
	NmC	56	17
	NmW	2	1
	NmX	40	12
	NmY	1	0
	Negative	221	68
Sp RDT results	Positive	28	9
	Negative	295	91

**Table 2 microorganisms-09-00832-t002:** Concordance of results obtained by RDT in PHCs and NRLs with results obtained by PCR in NRLs, and serogroup-specific performances, Burkina Faso and Niger, 2018–2019.

		PCR Negative/RDT Negative	PCR Positive/RDT Positive	PCR Negative/RDT Positive	PCR Positive/RDT Negative	Total	Sensitivity (%)	95% CI	Specificity (%)	95% CI	PPV (%)	95% CI	NPV (%)	95% CI
**RDT at health center**	All Nm	217	82	24	4	327	95	(89;99)	90	(86;94)	77	(68;85)	98	(95;100)
	NmA	318	0	9	0	327	NA		97	(95;99)	NA		100	(99;100)
	NmC	266	50	6	4	326	93	(82;98)	98	(95;99)	89	(78;96)	99	(96;100)
	NmW	324	0	2	0	326	NA		99	(98;100)	NA		100	(99;100)
	NmX	284	29	11	3	327	91	(75;98)	96	(93;98)	73	(56;85)	99	(97;100)
	NmY	324	0	1	0	325	NA		100	(98;100)	NA		100	(99;100)
	Sp	293	24	4	2	323	92	(75;99)	99	(97;100)	86	(67;96)	99	(98;100)
**RDT at NRL**	All Nm	108	28	6	2	144	93	(78;99)	95	(89;98)	82	(66;93)	98	(94;100)
	NmA	141	0	6	0	147	NA		96	(91;99)	NA		100	(97;100)
	NmC	127	16	2	1	146	94	(71;100)	98	(95;100)	89	(65;99)	99	(96;100)
	NmW	142	0	3	0	145	NA		98	(94;100)	NA		100	(97;100)
	NmX	132	12	0	1	145	92	(64;100)	100	(97;100)	100	(74;100)	99	(96;100)
	NmY	144	0	2	0	146	NA		99	(95;100)	NA		100	(98;100)
	Sp	138	8	0	1	147	89	(52;100)	100	(97;100)	100	(63;100)	99	(96;100)

RDT: rapid diagnostic test; PHC: primary health care center; NRL: national reference laboratory; PCR: polymerase chain reaction, 95% CI: 95% confidence intervel; PPV: positive predictive value; NPV: negative predictive value; Nm: Neisseria meningitidis; Sp: Streptococcus pneumoniae.

**Table 3 microorganisms-09-00832-t003:** Concordance of results obtained by RDT in PHCs with results obtained by PCR in NRLs, and serogroup specific performances, by year of lumbar puncture, Burkina Faso and Niger, 2018–2019.

		PCR Negative/RDT Negative	PCR Positive/RDT Positive	PCR Negative/RDT Positive	PCR Positive/RDT Negative	Total	Sensitivity (%)	95% CI	Specificity (%)	95% CI	PPV (%)	95% CI	NPV (%)	95% CI
2018	All Nm	80	16	17	1	114	94	(71;100)	83	(73;89)	49	(31;67)	99	(93;100)
	NmA	106	NA	8	NA	114	NA	NA	93	(87;97)	NA	NA	100	(97;100)
	NmC	110	4	NA	NA	114	100	(40;100)	100	(97;100)	100	(40;100)	100	(97;100)
	NmW	112	NA	1	NA	113	NA	NA	99	(95;100)	NA	NA	100	(97;100)
	NmX	93	12	8	1	114	92	(64;100)	92	(85;97)	60	(36;81)	99	(94;100)
	NmY	114	NA	NA	NA	114	NA	NA	100	(97;100)	NA	NA	100	(97;100)
	Sp	108	4	1		113	100	(40;100)	99	(95;100)	80	(28;100)	100	(97;100)
2019	All Nm	135	64	7	3	209	96	(88;99)	95	(90;98)	90	(81;96)	98	(94;100)
	NmA	208	NA	1	NA	209	NA	NA	100	(97;100)	NA	NA	100	(98;100)
	NmC	154	45	5	4	208	92	(80;98)	97	(93;99)	90	(78;97)	98	(94;99)
	NmW	208	NA	1	NA	209	NA	NA	100	(97;100)	NA	NA	100	(98;100)
	NmX	188	17	3	1	209	94	(73;100)	98	(96;100)	85	(62;97)	100	(97;100)
	NmY	207	NA	1	NA	208	NA	NA	100	(97;100)	NA	NA	100	(98;100)
	Sp	182	20	3	2	207	91	(71;99)	98	(95;100)	87	(66;97)	99	(96;100)

RDT: rapid diagnostic test; PHC: primary health care center; NRL: national reference laboratory; PCR: polymerase chain reaction; 95% CI: 95% confidence intervel; PPV: positive predictive value; NPV: negative predictive value; Nm: Neisseria meningitidis; Sp: Streptococcus pneumoniae.

**Table 4 microorganisms-09-00832-t004:** Concordance of results obtained by RDT in PHCs with results obtained by PCR in NRLs, and serogroup specific performances, by epidemic vs. non epidemic period, Burkina Faso and Niger, 2018–2019.

		PCR Negative/RDT Negative	PCR Positive/RDT Positive	PCR Negative/RDT Positive	PCR Positive/RDT Negative	Total	Sensitivity (%)	95% CI	Specificity (%)	95% CI	PPV (%)	95% CI	NPV (%)	95% CI
Epidemic period	All Nm	123	64	8	3	198	96	(88;99)	94	(88;97)	89	(79;95)	98	(93;100)
	NmA	197	NA	1	NA	198	NA	NA	100	(97;100)	NA	NA	100	(98;100)
	NmC	143	45	5	4	197	92	(80;98)	97	(92;99)	90	(78;97)	97	(93;99)
	NmW	197	NA	1	NA	198	NA	NA	100	(97;100)	NA	NA	100	(98;100)
	NmX	176	17	4	1	198	94	(73;100)	98	(94;99)	81	(58;95)	99	(97;100)
	NmY	196	NA	1	NA	197	NA	NA	100	(97;100)	NA	NA	100	(98;100)
	Sp	174	17	3	2	196	90	(67;99)	98	(95;100)	85	(62;97)	99	(96;100)
Non epidemic period	All Nm	94	18	16	1	129	95	(74;100)	86	(78;92)	53	(35;70)	99	(94;100)
	NmA	121	NA	8	NA	129	NA	NA	94	(88;97)	NA	NA	100	(97;100)
	NmC	123	5	1	NA	129	100	(48;100)	99	(96;100)	83	(36;100)	100	(97;100)
	NmW	127	NA	1	NA	128	NA	NA	99	(96;100)	NA	NA	100	(97;100)
	NmX	108	12	7	2	129	86	(57;98)	94	(88;98)	63	(38;84)	98	(94;100)
	NmY	128	NA	NA	NA	128	NA	NA	100	(97;100)	NA	NA	100	(97;100)
	Sp	119	7	1	NA	127	100	(59;100)	99	(95;100)	88	(47;100)	100	(97;100)

RDT: rapid diagnostic test; PHC: primary health care center; NRL: national reference laboratory; PCR: polymerase chain reaction; 95% CI: 95% confidence intervel; PPV: positive predictive value; NPV: negative predictive value; Nm: Neisseria meningitidis; Sp: Streptococcus pneumoniae.

**Table 5 microorganisms-09-00832-t005:** Concordance of the results of RDTs as interpreted live in the PHCs and by independent reading of photographs, Burkina Faso and Niger, 2018–2019.

	Test	Photo Reading Negative/RDT Negative	Photo Reading Positive/RDT Positive	Photo Reading Negative/RDT Positive	Photo Reading Positive/RDT Negative	Total	Concordance (%)	Kappa Coefficient
RDTs at health centers	All Nm	36	17	11	1	65	82	61
	NmA	64	2	6	1	73	90	32
	NmC	71	1	3	0	75	96	39
	NmW	68	1	0	4	73	95	32
	NmX	53	13	4	0	70	94	83
	NmY	71	0	0	0	71	100	NA
	Sp	66	5	1	1	73	97	82

RDT: rapid diagnostic test; PHC: primary health care center; Nm: Neisseria meningitidis; Sp: Streptococcus pneumoniae.

## Data Availability

Not applicable.

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
