# Peer review of "Field Evaluation of the Performance of Two Rapid Diagnostic Tests for Meningitis in Niger and Burkina Faso"

_microorganisms, 2021, doi:10.3390/microorganisms9040832_

Round 1
Reviewer 1 Report
This manuscript presents data investigating the results of lateral flow tests for Neisseria meningitidis and Streptococcus pneumoniae, called MeningoSpeed and PneumoSpeed, respectively. Importantly, these investigations were done ‘in the field’ at health centres in Burkina Faso and Niger with results from these lateral flow test compared to PCR tests from the national reference laboratories.
The abstract and introduction are well written, and clearly describe the content of the manuscript as well as the background context in which the research was done.
The methods that are presented are well described. It is commendable that the experimental design includes the photographing of the rapid detection tests at the public health clinics as they were being read and that these photographs were then blindly re-interpreted by independent readers.
The results and discussion are also well written and well presented.
Given the impact of the tests claimed in the discussion and conclusions, the manuscript needs to outline the process of using the MeningoSpeed and PneumoSpeed tests. This should include the steps involved and the equipment needed to successfully conduct a test, so that readers are aware of the skill and facilities that are needed at centres that may wish to implement these tests. This could perhaps be briefly included in the methods and then highlighted in the discussion with regards to feasibility.
There is an alignment issue with the text below Table 5.
Author Response
We thank the Reviewer for her/his positive opinion and for her/his feedback.
To meet the point of the Reviewer, we added (lines 105-110) a short description of the process of the tests in the ”Introduction” section. We further discussed in the ‘Discussion” section the advantage of such tests that do not require specialized staff, do not require additional equipment and can be performed within 15 minutes.
The alignment of the text was also fixed.
Reviewer 2 Report
The authors of this manuscript describe the performance of lateral flow devices to detect Neisseria meningitidis (serogroups A, C, W, X and Y) and Streptococuss pneumoniae. The trial was conducted over two years with samples from Niger and Burkina Faso in parallel with testing via Real-Time PCR. The study tested a total of 327 samples.
My thoughts are that at present there is little information on the field testing of these tests which makes this manuscript interesting reading to people working in this field. The sample size is appropriate for what it is try to achieve and was used in an active outbreak. The results of the questionnaire is helpful to build confidence in the tests. Overall, this is a well written manuscript and the study design has been thought through properly. However, I have the following concerns.
No declaration of conflict of interest. I know that this journal doesn't usually have this section but when evaluating a commercial test, I think that one needs to be present.
Consider moving Table 3 and 4 to supplementary section. They don't add much to the manuscript and can be referred to in the text.
Minor corrections:
Line 34: Although personal phrasing is acceptable, starting two consecutive sentences with the same word is not (both start with "We"). Please change either sentence.
Sentence starting line 91: This sentence is very long (six lines) and confusing. What does "first redeveloped" mean? Was it developed by one company and then redeveloped by multiple companies? Please break this long sentence down and make it clear.
Line 123: Is there a reference for the real-time PCR used that could be added so readers could check the method if they wanted.
Author Response
Reviewer 2
The authors of this manuscript describe the performance of lateral flow devices to detect Neisseria meningitidis (serogroups A, C, W, X and Y) and Streptococuss pneumoniae. The trial was conducted over two years with samples from Niger and Burkina Faso in parallel with testing via Real-Time PCR. The study tested a total of 327 samples.
My thoughts are that at present there is little information on the field testing of these tests which makes this manuscript interesting reading to people working in this field. The sample size is appropriate for what it is try to achieve and was used in an active outbreak. The results of the questionnaire is helpful to build confidence in the tests. Overall, this is a well written manuscript and the study design has been thought through properly. However, I have the following concerns.
We thank the Reviewer for her/his positive feedback.
No declaration of conflict of interest. I know that this journal doesn't usually have this section but when evaluating a commercial test, I think that one needs to be present.
Answer: Lines 138-140: We added the following statement “The producer of the tests (BioSpeedia Company) did not fund the trail, did not contribute to the design and did not participate in the analysis of the results. None of the authors is employed by that company”.
Consider moving Table 3 and 4 to supplementary section. They don't add much to the manuscript and can be referred to in the text.
Answer: Although these two tables provide details of the trial, we still feel that they are informative and we prefer keeping them in the text to facilitate their accessibility to Readers who may be interested by those details.
Minor corrections:
Line 34: Although personal phrasing is acceptable, starting two consecutive sentences with the same word is not (both start with "We"). Please change either sentence.
Answer: We changed the second sentence to “. The tests were performed in the cerebrospinal fluid of suspected meningitis cases from health centers in alert districts and compared to…”
Sentence starting line 91: This sentence is very long (six lines) and confusing. What does "first redeveloped" mean? Was it developed by one company and then redeveloped by multiple companies? Please break this long sentence down and make it clear.
Answer: We changed this sentence into two separate sentences. The first mentioned the first “academic development” by the Institut Pasteur of these tests. The second sentence described the transfer of the methods to the Company “BioSpeedia” for industrial development. (Lines 95-100 of the revised version)
“The rapid diagnostic tests (RDT) were first developed at the Institut Pasteur, Paris as immunochromatographic tests (13,14). They were thereafter transferred the BioSpeedia company who then developed MeningoSpeed and PneumoSpeed that can be used between 2°C and 30°C to diagnose, respectively, the five main meningococcal serogroups (A, C, W, X and Y) (15)”
Line 123: Is there a reference for the real-time PCR used that could be added so readers could check the method if they wanted.
Answer: We added the reference. This is the PCR that is recommended by the WHO “WHO. Laboratory methods for the diagnosis of meningitis caused by Neisseria meningitidis, Streptococcus pneumoniae, and Haemophilus influenzae, 2nd ed. ed.; WHO/IVB: Geneva, 2011”
Reviewer 3 Report
The paper from Rondy et al is a nice short straightforward test of two rapid tests for meningitis caused by meningococci (Nmen) and pneumococci (Spn) in real live situations (not controlled lab conditions) in African countries. The data are robust and the paper well-written. I have only a few trivial comments.
- Introduction - you would think that Nmen disease is still a burden in the Meningitis Belt, but the reality is that it has been tremendously reduced by MenA vaccination. the authors should just comment with some indication of rates per 100,000 for the whole belt before and after vaccination. The same for Spn - how many cases per 100,000 are seen?
- The introduction should include some more information on the components of the kits, without reading previous papers. Are these based on antibodies to serogroup polysaccharides? What is in the Spn test (where the world is your oyster as regards CPS?)?
- Comment - have these tests been used in patients with suspected sepsis?
- Limitations of the study - generally clear, but authors should mention that taking photographs does not necessarily provide quality imagery, and (taking covid tests as an example) bespoke 'readers' might provide more robust evaluations that readings by eye.
- Comment - line 189 - 191. is this correct that from LP and PCR result, the median times are between 18-22 days! Almost seems pointless doing them if you have to wait that long for a positive result. presumably patient are treated on clinical signs without recourse to confirmatory PCR.
Author Response
Reviewer 3
The paper from Rondy et al is a nice short straightforward test of two rapid tests for meningitis caused by meningococci (Nmen) and pneumococci (Spn) in real live situations (not controlled lab conditions) in African countries. The data are robust and the paper well-written. I have only a few trivial comments.
- Introduction - you would think that Nmen disease is still a burden in the Meningitis Belt, but the reality is that it has been tremendously reduced by MenA vaccination. the authors should just comment with some indication of rates per 100,000 for the whole belt before and after vaccination. The same for Spn - how many cases per 100,000 are seen?
Answer: Our main focus here is to discuss the variety of the Meningitis etiology, which is why we intentionally did not expand on Men A vaccine impact. In the introduction we slightly changed the following sentence to reflect the tremendous impact of Men A vaccination:
In 2019, following the massive reduction of Nm A with Men A vaccine rollout, Streptococcus pneumoniae (Sp) was found to be responsible for 40% of meningitis cases in countries in the region (8).
- The introduction should include some more information on the components of the kits, without reading previous papers. Are these based on antibodies to serogroup polysaccharides? What is in the Spn test (where the world is your oyster as regards CPS?)?
Answer: Lines 98-101, We added information on RDTs that are indeed based on antibodies against capsular polysaccharide for N. meningitidis serogroups A, C, W, Y and X. For Spn, the test is based on antibodies directed against cell wall polysaccharide C that is common to all Spn isolates.
- Comment - have these tests been used in patients with suspected sepsis?
Answer: Yes. Any patient complying with the WHO case definition of bacterial meningitis was eligible for this study. For some reasons, the footnote referring to the suspected case definition had disappeared from the reviewed manuscript.
It was added:
Sudden onset of fever and stiff neck or other meningeal signs (including bulging fontanelle for patients under 12 months).
- Limitations of the study - generally clear, but authors should mention that taking photographs does not necessarily provide quality imagery, and (taking covid tests as an example) bespoke 'readers' might provide more robust evaluations that readings by eye.
Answer: We do agree with the Reviewer. However, the test was introduced for the first time in field and we had no bespoke 'reader’s at the first stage of the work. In the future, automatic reader may provide more robust and objective solution.
- Comment - line 189 - 191. is this correct that from LP and PCR result, the median times are between 18-22 days! Almost seems pointless doing them if you have to wait that long for a positive result. presumably patient are treated on clinical signs without recourse to confirmatory PCR.
Answer: The Reviewer is right. Treatment is provided d of course. The point we wanted to underline her is that PCR cannot be used in remote villages where the epidemic may start. Samples need to be grouped and to be sent to regional centre to perform PCR. However, RDT can be used in these remote sites and can provide confirmatory results immediately.